# Age-Related Changes in Attentional Refocusing during Simulated Driving

**DOI:** 10.3390/brainsci10080530

**Published:** 2020-08-07

**Authors:** Eleanor Huizeling, Hongfang Wang, Carol Holland, Klaus Kessler

**Affiliations:** 1Aston Neuroscience Institute, Aston University, Birmingham B4 7ET, UK; h.wang26@aston.ac.uk; 2Aston Research Centre for Healthy Ageing, Aston University, Birmingham B4 7ET, UK; c.a.holland@lancaster.ac.uk

**Keywords:** ageing, simulated driving, attention, switching costs, neural oscillations, Neuro-VR

## Abstract

We recently reported that refocusing attention between temporal and spatial tasks becomes more difficult with increasing age, which could impair daily activities such as driving (Callaghan et al., 2017). Here, we investigated the extent to which difficulties in refocusing attention extend to naturalistic settings such as simulated driving. A total of 118 participants in five age groups (18–30; 40–49; 50–59; 60–69; 70–91 years) were compared during continuous simulated driving, where they repeatedly switched from braking due to traffic ahead (a spatially focal yet temporally complex task) to reading a motorway road sign (a spatially more distributed task). Sequential-Task (switching) performance was compared to Single-Task performance (road sign only) to calculate age-related switch-costs. Electroencephalography was recorded in 34 participants (17 in the 18–30 and 17 in the 60+ years groups) to explore age-related changes in the neural oscillatory signatures of refocusing attention while driving. We indeed observed age-related impairments in attentional refocusing, evidenced by increased switch-costs in response times and by deficient modulation of theta and alpha frequencies. Our findings highlight virtual reality (VR) and Neuro-VR as important methodologies for future psychological and gerontological research.

## 1. Introduction

The population of older drivers is rapidly increasing. Identifying age-related changes in cognition that impair driving performance is important to ensure that older adults can continue to drive safely. Although older adults have an overall reduced crash risk compared to young drivers, statistics show that they present a disproportionate risk of at-fault collisions at intersections and collisions caused by a failure to give-way, or to notice other objects, stop signs or signals [1,2,3,4]. Older drivers typically report an increased subjective difficulty in processing signs in time [5]. Overall, the evidence suggests that older drivers’ collisions are caused by failures in selective attention and switching [6], and that driving skills later in life may be impaired by declines in switching or refocusing attention.

In our recent work [7], we found an age-related decline in the ability to refocus attention from attending to temporally changing events to spatially distributed stimuli. These findings are in line with more general declines in attention with increased age [8,9,10,11,12,13,14,15]. Furthermore, Choi et al. [16] recently reported that reduced attentional control correlates with crash risks during simulated driving, specifically in situations that require a fast resolution of conflicts among competing tasks. Difficulties in switching between temporal events and spatially distributed stimuli [7] could relate to difficulties in switching from attending to the dynamic changes in traffic on the road ahead to attending to road signs and other surrounding objects and events.

In addition to attentional refocusing, the work by Callaghan, Holland and Kessler [7] included an element of goal switching. Participants switched from the goal of identifying a number in a stream of letters to the goal of identifying a letter in a visual search display. Older adults have been shown to find task switching particularly difficult when required to maintain more than one task goal [17]. In the context of driving, one is required to maintain several task goals at once, for example vehicle control, route finding and monitoring traffic, pedestrians and bicycles. Goal switching between different tasks could, therefore, be substantially more challenging for older adults while driving in a dynamically evolving scenario (e.g., [18]).

In the current study, age groups were compared on their ability to switch from allocating attention to dynamic events in time, where participants must attend to the fast changing traffic in front of them (spatially focal), to distributing attention spatially, in order to complete a visual search of a road sign (the target city name “Birmingham” was embedded within 11 other city names, see Figure 1). Participants drove on a simulated dual-carriageway. To maintain immersion, driving events ran on continuously and dynamically from one another, clustering into event pairs or single events, depending on “trial” condition. In Sequential-Task Switch trials, the “road sign visual search” task (2nd task) was preceded by a “braking event” task (1st task), where participants were required to brake in response to a car suddenly driving in front of them from the over-taking lane and braking. Shortly after the braking event, participants were required to refocus their attention spatially in order to complete the “road sign visual search” task. In “Single-Task” trials the “road sign visual search” task (only task) was carried out without a preceding “braking event” task. It was hypothesised that response times (RTs) in response to the road sign would be slower when the sign was preceded by a braking event in Sequential-Task trials, compared to when the sign appeared without a preceding braking event task in Single-Task trials. We refer to this slowing of RTs in Sequential-Task compared to Single-Task trials as Sequential-Task Costs. Crucially, based on our prior work and the existing literature [7,19,20], we hypothesised that there would be an age-related decline in refocusing attention from the temporal to the spatial task while driving, reflected in a greater proportional increase in RTs in Sequential-Task compared to Single-Task trials (Sequential-Task Costs).

Electroencephalography (EEG) was recorded in a subset of 17 participants aged 18–30 and 17 aged 60+ years as a proof-of-principle pilot investigation. Although previous literature has successfully recorded EEG in driving simulator environments [21,22,23], many studies investigated the effects of fatigue on EEG signals [22,24,25,26] and only a very limited number of studies have investigated task-related modulations of oscillatory signatures [27,28,29]. To our knowledge, no study to date has investigated changes to task-related oscillatory signatures in older drivers during naturalistic, simulated driving. We believe this to be an important gap to fill, because oscillatory signatures at alpha, beta and theta frequency have been linked with a variety of attentional and executive functions and their age-related decline, which could be crucial for complex daily activities such as driving.

Based on an extensive body of research on target processing, where increased alpha power is typically thought to reflect inhibition, whereas an alpha desynchronization is thought to reflect enhanced attention [30,31,32,33,34,35,36], we expected task-related (Single- vs. Sequential-Task) alpha modulations in relation to the onset of the braking event and subsequently to the appearance of the road sign. Additionally, it was hypothesised that EEG theta power would be modulated by Single- vs. Sequential-Task settings, in accord with previous research linking such modulations with variations in the level of top-down guided attentional control and target processing [37,38,39,40,41].

Age-related changes in alpha and theta oscillations have been linked to poorer attention and executive control [42,43,44,45,46,47]. We, therefore, hypothesised that the older group would show weaker task-related theta and alpha power modulation compared to younger adults. Previous research suggests that age differences in oscillatory patterns during selective attention originate from a fronto-parietal network that has been associated with attentional control [11,48,49,50,51,52]. Indeed, in a magnetoencephalography (MEG) study, Wiesman and Wilson [42] found age effects in oscillatory signatures in a fronto-parietal network during a visuospatial attention task.

However, older drivers may be able to increase the amount of cognitive resources applied during a critical task such as driving, especially if they still regularly engage in this activity, which would be consistent with reports of compensatory recruitment of top-down processes in older age [51,53,54] and would then be reflected in increased and wider-spread task-related modulation of alpha and theta oscillations in the older compared to the younger group.

## 2. Methods

### 2.1. Participants

Data from one-hundred and eighteen participants in five age groups (18–30, 40–49, 50–59, 60–69, and 70–91 years) were collected and analysed. A G *Power calculation revealed that a sample of 110 participants was required for detecting a medium effect size (f = 0.35) with 0.9 statistical power. An additional ten participants did not complete the experiment due to simulator sickness, including four participants from the 50–59 years group and two participants from each of the 40–49, 60–69, and 70–91 years groups. All participants had a full driving license, had experience driving in the United Kingdom (UK) and had driven within the last year. Consistent with Callaghan, Holland and Kessler [7], the 18–30 years group were used as a comparison group for age-related cognitive changes for all other groups. The 40–49 and 50–59 years groups provided middle-aged comparison groups for the 60–69 and 70–91 years groups. Age ranges were selected to assess performance not only in older adults, but also in middle age. Participants with photosensitive epilepsy or uncorrected visual impairments were excluded from participation, in addition to those who scored equal to or less than the cut-off for possible cognitive impairment of 87 on the cognitive assessment used, the Addenbrookes Cognitive Examination 3 (ACE-3 [55]).

Participants in the 18–30, 40–49 and 50–59 years groups were recruited from Aston University staff and students and the community. Participants aged over 60 years were recruited from the Aston Research Centre for Healthy Ageing (ARCHA) participation panel. Participants received course credits (students) or a standard fee towards their travel expenses. All participants provided written informed consent before participating. The research was approved by Aston University Research Ethics Committee (approval code #897) and complied with the Declaration of Helsinki.

Two participants from the 70–91 years group scored equal to or lower than the cut-off of 87 on the ACE-3 [55] and were therefore excluded from further analyses. The remaining participants’ demographics are presented in Table 1.

### 2.2. Materials and Procedures

#### 2.2.1. Driving Simulator

Participants completed a driving simulator task where they switched from allocating their attention to events changing in time (i.e., temporal attention), as they attended to the fast changing traffic on the road ahead, to distributing attention spatially, in order to complete a visual search of a road sign.

An example of the experimental setup is displayed in Figure 1. Participants were seated comfortably in an adjustable GT Omega Art Racing Simulator Cockpit (RS6 seat), complete with a Logitech G27 Force feedback wheel and pedal set, which incorporated a steering wheel, gear stick, clutch, brake and accelerator pedals. The indicators were paddles to the left and right of the steering wheel that participants could pull towards them to turn on and off. A manual gearstick was to the left of the participant and was programmed to go up to 5th gear. Driving simulator software STISIM Drive™ by Systems Technology Inc. (kernel build: Build 2.10.09) was used to record driving simulator data and to render the driving simulations, which were projected at a resolution of 1280 × 1024 pixels onto three 1.30 × 2.27 m projection screens at a refresh rate of 75 Hz. Data were sampled at a frequency of 60 Hz. The central projection screen was positioned facing the driving seat 2.20 m away from the participant. To fabricate the perception of movement through 3D space, the two peripheral projection screens were positioned adjacent to the central screen, rotated 40 degrees away from the central screen towards the driving seat. The projection included a dashboard which contained a speedometer displaying miles per hour (mph) and a rev counter displaying revolutions per minute. The driving seat was surrounded by a speaker system through which engine and braking sound-effects were produced.

Participants drove on a simulated dual-carriageway. To maintain immersion in the driving simulation study “trials” ran on continuously from one another, where certain events, which occurred either in pairs or in isolation within the ongoing driving scenario, were considered as trials. In “Sequential-Task” trials, participants were required to brake in response to a car suddenly pulling in front of them from the over-taking lane. Participants were instructed to brake as quickly as possible and RTs to brake initiation were recorded (see Appendix A). Shortly after this “braking event”, participants were required to refocus their attention spatially to complete a visual search of a road sign, which appeared in front of the driver after they had travelled either another 3.04 m (Immediate Switch condition: ~0.14 s, assuming they are driving at the speed limit) or another 30.48 m (Delayed Switch condition: ~1.42 s, assuming they are driving at the speed limit) after the braking event. The Delayed Switch condition was included to prevent participants predicting exactly when the road sign would appear. In “Single-Task” trials the sign appeared without a preceding braking event. In Sequential-Task trials, the delay between the braking event and the road sign was specified in distance, resulting in the time delay between the braking event and the road sign varying with participants’ driving speeds. Note that the Single-Task condition still involved an attention switching element, where participants attended to the road ahead before switching to attend to the road sign. However, Sequential-Task trials were expected to require a heightened effort, both to refocus attention due to a greater enhancement of attention towards temporal events (i.e., towards braking), and to switch from the task goals of braking in response to the braking event to the task goals of indicating in response to the road sign.

An example of the road sign is displayed in Figure 1 (Panel B, top). When the road sign appeared, participants were required to identify the location of the target word “Birmingham” and indicate left if it was in the left column, to signal that they would exit the dual-carriageway, and right if it was in either the middle column or the right hand column to signal that they would stay on the dual-carriageway. Participants were instructed to indicate as quickly and accurately as possible and indicator RTs and accuracy were recorded. The speed at which participants were travelling when the sign appeared was also recorded (see Appendix A). Participants were instructed that the speed limit was 70 mph (consistent with the speed limit on UK dual-carriageways). The maximum speed that participants could drive was programmed at 80 mph.

The city name “Birmingham” was chosen as a visual search target word, as this was the city where the study took place and was therefore familiar to all participants. The target was embedded among 11 distractor names. The stimuli remained the same on all trials and only the order of the stimuli on the sign differed across trials. Distractor stimuli were UK city names. To avoid advantaging participants who were more familiar with certain roads than others, the order of the names on the sign was random. Participants were informed that signs did not represent realistic road signs with regards to the order of names. On 50% of trials the target was in the left column and on 50% of the trials the target was in one of the two right-hand columns, so that 50% of trials required a left-hand indicator response and 50% of trials required a right-hand indicator response (see above for details). There were 36 trials in total and trials were divided into three blocks of 12 trials to provide opportunities for breaks. There were 12 trials in each condition that were pseudorandomised throughout the three blocks. The order of trials in each block can be found in Appendix A. Each block took approximately 15 min, depending on the speed at which participants were driving. Due to the length of the experiment (45 min), it was not plausible to increase the number of trials due to concerns about the effects of fatigue on the performance [56]. Fatigue has been shown to effect neural responses recorded with EEG in the driving simulator [24]. Furthermore, a small number of trials per condition maintains the naturalistic nature of the experiment, by preventing over-repetition and minimising the chance to develop artificial strategies that would not be present when responding to surprising events in everyday on-road driving. There are examples across multiple cognitive domains where increasing the number of trials weakens power due to effects of learning throughout the experiment, e.g., [57]. Minimising the number of trials prevents such learning from taking place and better reflects everyday driving performance.

Prior to beginning the task participants took part in two practice driving scenarios. In the first scenario, participants were given the opportunity to familiarise themselves with the controls of the driving simulator while driving around a virtual town. Participants continued driving in the town scenario until they felt confident with the controls, particularly with changing gear, steering and braking. The aim of the second practice scenario was to familiarise participants with the task instructions. Participants completed six practice trials of the task; however, trials differed from experimental trials as they contained no traffic on the road and so there were no braking events and no Sequential-Task switching element to the task.

#### 2.2.2. EEG Acquisition

From the 118 participants, we recorded EEG in 17 participants aged 18–30 years (mean = 22.88 years, SD = 4.05) and in 17 participants aged 60+ years (mean = 70.12 years, SD = 5.20) while they completed the driving simulator task. EEG was recorded with a 64 channel eego™ sports mobile EEG system (ANT Neuro, Enschede, The Netherlands) and digitised at a sampling rate of 500 Hz. Sensors were Ag/AgCl electrodes arranged in accordance with the International 10–10 system. Electrode CPz was taken as an online reference electrode and the ground electrode was positioned at AFz. Participants were instructed to keep their face as relaxed as possible throughout the recording and to keep their head movements to the minimum necessary while driving to minimise muscle artefacts.

## 3. Data Analysis

### 3.1. Driving Simulator Task RTs and Accuracy

In the driving simulator task, participants’ median indicator RTs on trials where they had both braked successfully in the braking event and indicated correctly in response to the road sign were extracted from raw driving simulator outputs. The proportion of correct indicator responses, braking responses and participants’ braking RTs (see Appendix A) and median driving speeds (mph; Appendix A) when passing the road sign were also recorded.

Differences in median indicator RTs between event conditions and age groups were analysed in a 3 × 5 mixed ANOVA, where event condition (Immediate Switch/Delayed Switch/Single-Task) was a within-subjects factor and age group (18–30/40–49/50–59/60–69/70–91 years) was a between-subjects factor. Multiple comparisons were corrected for with Bonferroni correction.

The data were expected to violate assumptions of equality of variance due to increases in inter-individual variability with age [58,59], yet, there is evidence to support that the ANOVA is robust to violations to homogeneity of variance [60,61]. Where Mauchly’s Test of Sphericity was significant, indicating that the assumption of sphericity had been violated, Greenhouse–Geisser corrected statistics were reported.

To interpret the age group × event condition interaction that was identified in the indicator RT ANOVA, percentage differences from the Single-Task condition to each of the Sequential-Task conditions were calculated as measures of “Sequential-Task Costs” for each individual, and independent t-tests were implemented to compare age groups’ Sequential-Task Costs. As interaction effects were already shown to be statistically significant in the ANOVA, Restricted Fisher’s Least Significant Difference test was applied and corrections for multiple comparisons were not conducted [62]. Where Levene’s test for equality of variance was significant (*p* < 0.05), “Equality of variance not assumed” statistics were reported.

### 3.2. EEG Analysis

#### 3.2.1. Preprocessing

EEG data were read into the Matlab2017a^®^ toolbox Fieldtrip version 20151004 and analysed with version 20161031 [63], bandpass filtered between 0.5–36.0 Hz and epoched from −7.00–3.00 s, where 0.00 s corresponded to the onset of the road sign. Trials were visually inspected for artefacts and trials with large artefacts were removed in addition to trials where participants failed to brake successfully in the braking event or indicate correctly in response to the road sign (Single-Task mean = 1.35 trials, SD = 1.25; Immediate Switch mean = 0.62 trials, SD = 0.78; Delayed Switch mean = 2.50 trials, SD = 0.86). Prior to analysis, independent component analysis was implemented and components with eye-blink or heartbeat signatures were omitted.

#### 3.2.2. EEG Sensor Level Analysis

Noisy sensors were interpolated with the averaged signal from neighbouring electrodes. Time-frequency analysis was carried out by applying a Hanning taper from 2–30 Hz (for every 1 Hz), with five cycles per time-window in steps of 50 ms. For each participant, trials were averaged within each condition (Single-Task/Immediate Switch/Delayed Switch). No baseline correction was applied due to potential group differences in baseline cognitive states. Instead, conditions were compared directly.

#### 3.2.3. Exploratory EEG Source Analysis

In order to explore possible cortical sources for the oscillatory effects that were evident at the sensor-level, and to explore oscillatory signatures while using spatial filtering to suppress noise external to the cortex, sources of theta (3–7 Hz; 0.00–0.80 s) and alpha (8–12 Hz; 1.0–2.0 s) oscillations were localised with exact Low-Resolution Electromagnetic Tomography (eLORETA; Pascual-Marqui, 2007), which has been shown to be less susceptible to noise than LORETA [64,65]. Time-frequency tiles were selected based on the average of time-frequency representations (TFRs) across all conditions and groups (see Appendix A), in order to analyse possible cortical origins of sensor level effects after the onset of the road sign. Note that the TFR in Appendix A displays power difference in relation to a baseline that was not used in any statistical analysis, collapsed across conditions and groups, and so does not constitute as “double-dipping”. Noisy electrodes were excluded prior to re-referencing data to the average of all remaining electrodes. Data were bandpass filtered and epoched to selected time-frequency tiles (detailed above). To avoid spectral leakage of the theta response (e.g. Appendix A), a time window of 1.00–2.00 s was selected for alpha source power analysis. A generic Boundary Element Method head-model was created from a template T1 weighted MRI. Head-models were normalised to MNI space (Montreal Neurological Institute template). Consistent with [66], source model voxels were 5 mm in size to improve the fit near the surface. Covariance matrices (computed for data pooled across conditions) were combined with estimated leadfields to produce common spatial filters. These spatial filters were subsequently applied to data from each condition separately.

#### 3.2.4. Statistical Analysis

The same statistical analysis procedure was followed for sensor and source level analysis in order to explore consistent effects across participants. Two-tailed dependent t-tests were carried out to compare each of the Sequential-Task conditions (Immediate Switch/Delayed Switch) with the Single-Task condition separately for each age group. Multiple comparisons were corrected for with non-parametric cluster permutations [67,68], with 2000 permutations (cluster alpha = *p* < 0.05). Second level analysis was carried out by comparing Sequential-Task Costs at the group level [69,70]. For each participant, the Single-Task condition was subtracted from each of the Sequential-Task conditions separately. These differences were entered into two-tailed independent cluster permutation t-tests (2000 permutations; cluster alpha = *p* < 0.05) to compare age groups. This statistical approach is consistent with recommendations on the Fieldtrip website “How to test an interaction effect using cluster-based permutation tests?” [71] and has been implemented in previous work [69,70].

## 4. Results

### 4.1. Driving Simulator Task

All age groups achieved greater than 98% accuracy to respond to the braking event and greater than 95% accuracy to respond to the road sign. No further analysis was conducted on accuracy data.

#### Indicator RTs

Group means of participants’ median indicator RTs in response to the road sign visual search are presented in Figure 2. (Braking RTs are displayed in Appendix A; alongside driving speeds when passing the road sign in Appendix A).

To investigate the hypothesis that Sequential-Task Costs would be greater in older compared to younger groups, a 3 × 5 (event condition × age group) ANOVA was conducted on participants’ indicator RTs. There was a significant main effect of age (*F* (4, 111) = 25.64, *p* < 0.001, η^2^*_p_* = 0.48) and event condition (*F* (1.66, 185.04) = 57.42, *p* < 0.001, η^2^*_p_* = 0.34) on indicator RTs and a significant age × event condition interaction (*F* (6.67, 185.04) = 3.74, *p* = 0.001, η^2^*_p_* = 0.12). The main effect of age resulted from significantly faster indicator RTs in the 18–30 years group compared to the 50–59 (*p* = 0.004), 60–69 (*p* < 0.001) and 70–91 (*p* < 0.001) years groups, and significantly slower indicator RTs in the 70–91 years group compared to all other groups (*p* < 0.001). There were no other significant age group differences in indicator RTs (*p* > 0.10). The main effect of event condition resulted from significantly faster indicator RTs in the Single-Task condition compared to both Sequential-Task Switch conditions (*p* < 0.001) and faster RTs in the Immediate Switch condition compared to the Delayed Switch condition (*p* < 0.001).

To investigate the hypothesis that age-related increases in Sequential-Task Costs would be seen while driving, the interaction between age and task condition was further explored. Each participant’s percentage increase in RTs from the Single-Task condition to the Immediate Switch condition (Immediate Sequential-Task Costs) and from the Single-Task condition to the Delayed Switch condition (Delayed Sequential-Task Costs) were calculated as measures of Sequential-Task Costs. Sequential-Task Costs were entered into independent t-tests to compare groups. T-tests were conducted to interpret the significant age group × event condition interaction and multiple comparisons were not corrected for (see Methods Section 3.1 and [62]). The means and SDs of each group’s Immediate and Delayed Sequential-Task Costs are presented in Table 2.

There were significantly higher Immediate Sequential-Task Costs in the 70–91 years group compared to the 18–30 years group (*t* (52) = −2.54, *p* = 0.014). Higher Immediate Sequential-Task Costs in the 40–49 years group compared to the 18–30 years group did not reach significance (*t* (52) = −1.75, *p* = 0.087). There were no other significant age group differences in Immediate Sequential-Task Costs between any age group (*p* > 0.10).

Compared to the 18–30 years group, there were significantly higher Delayed Sequential-Task Costs in the 50–59 (*t* (52) = −2.74, *p* = 0.008), 60–69 (*t* (54) = −2.48, *p* = 0.016) and 70–91 (*t* (27.95) = −2.22, *p* = 0.035) years groups. No other significant age group differences in Delayed Sequential-Task Costs were found between any age group (*p* > 0.10).

### 4.2. EEG

Group means of EEG participants’ median indicator RTs in response to the road sign visual search are presented in Figure 2 panel B. EEG participants’ RTs were included in the statistical analysis outlined in Section 4.1.

#### 4.2.1. Sensor-Level TFRs

The TFRs in Figure 3 display a theta response shortly after the road sign appears, which is consistent with top-down guided attentional control and target processing [37,38,39,40,41]. In the Sequential-Task conditions, there was an earlier theta increase that reflects processing of the braking event, which is initiated approximately −0.90 s relative to the onset of the visual search in the Immediate Switch condition and −2.00 s in the Delayed Switch condition. As the vehicle did not brake in front of the participant until either ~0.14 s prior to the onset of the sign in the Immediate Switch condition or ~1.42 s prior to the onset of the sign in the Delayed Switch condition, the earlier onset of theta modulation likely reflects detection of and attention to the vehicle approaching.

Figure 3 also illustrates an alpha power decrease in a late time window (starting around 0.50 s) in response to the onset of the road sign, consistent with enhanced attention to significant stimuli [30,31,32,33,34,35,36].

An early beta (15–25 Hz) decrease is also evident, which was initiated in response to the braking event and maintained throughout the visual search of the road sign. This could reflect enhanced attention or motor preparation as participants learned that a road sign would follow the braking event [72,73,74,75,76,77]. It appears that this is greater in the older compared to younger group. However, no further analysis was conducted on this, due to concerns about interference from muscle artefacts.

The naturalistic setting of the experiment meant that there was no suitable baseline period to statistically compare relative changes in power in response to task events that appear to be present in Figure 3. Instead, conditions were contrasted directly to compare power across conditions and age groups. To test the hypotheses that there would be age-related decreases in task-related theta and alpha modulation, frequency bands for theta (3–7 Hz) and alpha (8–12 Hz) were selected for further analysis.

#### 4.2.2. Single-Task vs. Sequential-Task Conditions

Figure 4 presents theta (3–7 Hz) and alpha (8–12 Hz) EEG power sensor level effects when contrasting Immediate Switch and Single-Task conditions in each age group (Panels A and C) and when contrasting Delayed Switch and Single-Task conditions in each age group (Panels B and D). No significant effects were found when investigating the event condition × age interactions (*p* > 0.10).

The 18–30 years group but not the 60+ years group displayed significantly higher sensor level alpha power in the Single-Task compared to Sequential-Task conditions (in both Delayed and Immediate Switch conditions). Group comparisons of differences were not significant (*p* > 0.10).

The younger but not the older group displayed significantly higher sensor level theta power in the Single-Task condition compared to both the Immediate Sequential-Task and Delayed Sequential-Task conditions, the cluster for which peaked after the road sign onset. The 60+ years group additionally showed weaker theta power in the Single-Task compared to Sequential-Task Immediate Switch condition in a time window preceding the road sign onset. This likely reflects increased processing in response to the temporal braking event before the onset of the sign in the Immediate Switch condition. Group comparisons of differences were not significant (*p* > 0.10).

#### 4.2.3. Exploratory Source Analysis

As described in Methods, we employed eLORETA to explore the plausible cortical origins of the condition effects found at sensor level (shown in Figure 4) in the period right after the onset of the road sign, and to explore the possible oscillatory signatures while suppressing noise through spatial filtering. The results of these exploratory source analyses could be used as a basis for future confirmatory research.

Consistent with the sensor level analysis, Figure 5 displays higher frontal theta and alpha source power in the Single-Task compared to Sequential-Task conditions in the younger group (Panels A–D, left column), lower posterior theta power in the Single-Task compared to Immediate Switch condition in the older group (Panel C, middle column), and higher frontal alpha power in the Single-Task compared to Delayed Switch condition in the older group (Panel D, middle column). Source analysis highlighted additional effects in the older group after road sign onset (Panels B–C, middle column) and at second level (group comparison) analysis (right column) that were not evident in the sensor level analysis, providing preliminary evidence that spatial filtering may be a useful tool to increase sensitivity in noisy, naturalistic environments. The results presented in Figure 5 suggest that the lower posterior theta power and higher frontal alpha power effects that occur before road sign onset in the older group, evident in Figure 4A,D, may actually have been present in the Single-Task compared to both Sequential-Task conditions, rather than in only the Immediate or Delayed Switch conditions, as Figure 4B,C (right column) would suggest. However, source analysis focused on the period after the onset of the sign, and thus, did not cover the period before onset, where the older group had displayed stronger theta in Sequential-Task conditions compared to Single-Task (Figure 3, bottom and Figure 4A, right).

## 5. Discussion

The aim of the study was to investigate whether age-related difficulties in switching from a temporal to a spatial attention task [7] can also be observed in a more naturalistic and ecologically valid setting, i.e., during simulated driving, where older and younger drivers were required to switch from a spatially focal yet temporally complex task (braking due to traffic ahead) to a spatially more distributed task (reading a motorway road sign). EEG was recorded in a subset of participants (17 younger and 17 older drivers) while they completed the driving simulator task, permitting the investigation of oscillatory signatures at theta and alpha frequencies. Our primary hypothesis, that there would be greater Switch Costs in older compared to younger age groups, was supported.

### 5.1. Response Times

Consistent with hypotheses, RTs to indicate were slower with increased age, in line with general age-related slowing of RTs [8,9,10,11,12]. As hypothesised, RTs to complete the visual search of the road sign were significantly slower in the two Sequential-Task conditions compared to the Single-Task condition, reflecting costs of switching from responding to a traffic event ahead (braking) to responding to a spatially distributed motorway road sign (signalling). In other words, it was more difficult to broaden the focus of attention to distribute attention spatially (to an overhead motorway road sign, in order to find a target city name among distractors) when attention mechanisms were already engaged in responding to a temporally complex event (the car braking in front) in the Sequential-Task conditions.

RTs in the Delayed Switch condition were significantly slower than in the Immediate Switch condition. This was unexpected, as faster RTs were predicted when participants had more time to refocus their attention spatially after the braking event. In our previous laboratory-based experiment [7], participants were faster on the visual search task when they could prepare to switch sooner. It is likely that participants learned that the road sign followed the braking event. After initially being a distraction, the braking event may have served as a cue for the motorway sign. There is a significant body of literature that demonstrates that temporal orienting of attention is enhanced at shorter (compared to longer) time intervals between a cue and a target, and RTs to detect a target decrease if the time of stimulus onset is predictable [50,78,79,80]. When the distance between the braking event and the road sign onset is farther in the Delayed Switch condition, the variability in the time of the road sign onset increases, due to variability in the speed that participants are driving (see also Appendix A). Increased RTs in the Delayed Switch condition compared to the Immediate Switch condition were therefore likely due to variability in the onset time of the road sign (making it less predictable), combined with a longer time period between the braking event and the road sign onset. There was no such variability in the onset time of the visual search task in Callaghan, Holland and Kessler [7], and so RTs were faster when participants had more time to prepare to switch. In addition, it could be that at long delays, participants focused their attention back on the road traffic and therefore had to re-focus attention spatially when the sign appeared, which shares some resemblance with the classic Inhibition-of-Return effect [81,82]. However, possible explanations of the cognitive mechanisms underlying longer RTs in the Delayed compared to Immediate Switch condition are merely speculative and would benefit from further research that could include eye-tracking, for instance. Such considerations also highlight the complex issues that have to be considered in dynamic naturalistic settings (e.g., [18]).

The hypothesis that there would be an age-related decline in switching attention while driving was supported. There was a greater increase in RTs from the Single-Task condition to the Immediate Switch condition in the 40–49 and 70–91 years groups compared to the 18–30 years group (although this did not reach significance in the 40–49 years group). Greater Delayed Sequential-Task Costs were found in the 50–59, 60–69 and 70–91 years groups compared to the 18–30 years group. These findings demonstrate that age-related declines in refocusing attention between tasks are not only observed in a standard computer-based paradigm [7] but are also present in more naturalistic settings, such as simulated driving.

Findings of significantly increased Delayed Sequential-Task Costs but not Immediate Sequential-Task Costs in the 50–59 and 60–69 years groups signifies that these age groups find switching easier when the onset time of the stimulus is after a short delay (~0.14 s) and predictable, but are more impaired when the time of stimulus onset is after a longer delay (~1.42 ms) and ambiguous—which is arguably more typical for real-life driving. These findings are in line with older age groups relying more on top-down guidance to control attention, such as implementing the use of temporal cues [83,84,85], as well as evidence towards impaired anticipatory attention mechanisms [86,87,88]. Furthermore, there is evidence to suggest that preparatory processes during task switching function well in older age and that some performance differences may result from age-related changes in the strategies employed for task implementation [89]. It is also important to highlight that older drivers were driving more slowly after braking, at the point of road sign onset, than the youngest group (Appendix A). This resulted in a slightly longer delay of road sign onset in older drivers, which may have precluded increased Immediate Sequential-Task Costs in the 50–59 and 60–69 years groups. In contrast, with a longer and even more speed-dependent delay of road sign onset in the Delayed Switch condition, attention may have been fully re-directed to the road traffic, thereby preventing any benefit from temporal cueing strategies used to anticipate the road sign. Finally, a contributing factor for the lack of significantly higher Immediate Sequential-Task Costs in the 50–59 and 60–69 years groups could be age-related increased variability in RTs, generally, and in Sequential-Task Costs more specifically.

In contrast to the 50–59 and 60–69 years groups, participants aged 40–49 years displayed higher Immediate Sequential-Task Costs but no difference in Delayed Sequential-Task Costs compared to participants aged 18–30 years. Note that Figure 2 displays little difference between Immediate and Delayed Switch RTs in the 40–49 years group, and so the discrepancies in age group differences between Immediate and Delayed Sequential-Task Costs are partly driven by higher Delayed compared to Immediate Sequential-Task Costs in the 18–30 years group (evident in Table 2).

### 5.2. EEG

The pattern of RT results was replicated in principle for the subset of 34 participants (17 younger and 17 older participants) for which EEG was recorded during simulated driving (Figure 2). EEG offered an online indicator of brain responses in relation to dynamically unravelling traffic events on-screen.

On inspection of Figure 3 (also Appendix A), theta power appeared to increase shortly after the road sign onset, consistent with the notion of theta involvement in top-down guided attentional control and visual search processing [37,38,39,40,41]. A later alpha desynchronization was apparent after road sign onset, likely related to increased attention to the road sign [30,31,32,33,34,35,36]. An early beta decrease was also seen, which was initiated in response to the braking event, and was maintained throughout the visual search of the road sign. It is likely that this reflects enhanced attention or motor preparation as participants learned that a road sign (requiring a response) would follow the braking event [72,73,74,75,76,77]. However, no further analysis was conducted on beta, due to concerns of interference from muscle artefacts due to active driving movements. Appendix A demonstrates that this was not a problem for theta and alpha frequencies. Furthermore, relative changes in power in response to task events, which appear to be present in Figure 3, were not tested statistically, as the naturalistic, dynamic and continuous nature of the experimental setup meant there was no suitable baseline period. Instead, conditions were compared directly.

The hypothesis that there would be a weaker theta response in the older compared to younger group was supported. In a time window after the onset of the road sign, the 18–30 years group displayed higher theta in the Single-Task condition compared to each of the Sequential-Task conditions. In contrast, the 60+ years group showed no significant difference between Single-Task and Sequential-Task conditions in sensor level theta power in response to the road sign. Interestingly, stronger theta power was observed for the older group before the onset of the road sign in the Immediate Sequential-Task condition (compared to the Single-Task condition), which was not observed for the younger group, and which could indicate a greater use of top-down, cue-based (braking event) strategies that has previously been reported in older adults [83,84,85]. However, when directly compared, group differences did not reach significance at sensor level. In contrast, group comparisons in source space indicated reduced posterior theta in older drivers (Figure 5) in response to the road sign, which reached statistical significance in the Delayed Switch contrast. This posterior reduction is in accord with recent MEG findings [90] using Callaghan, Holland and Kessler’s [7] rapid serial visual presentation (RSVP) – visual search paradigm.

Greater theta power in the Single-Task condition compared to Sequential-Task conditions is an interesting finding. It could be that there are fewer attentional resources available to process the sign in the Sequential-Task conditions after attending to the braking event. Alternatively, higher theta power in the Single-Task condition could reflect enhanced cognitive effort to attend to the road sign when driving at high speeds, in contrast to when participants brake shortly before the road sign onset in response to the braking event in the Sequential-Task conditions. If the latter hypothesis is correct, the theta power observed in the younger group could reflect enhanced attentional resources directed to the road sign (more so in the Single-Task compared to Sequential-Task condition), which contrasts from the older group, who drive at slower speeds to read the road sign. In addition, only the older drivers increased their theta after the braking event in anticipation of the road sign (Figure 3, bottom; Figure 4A, right), potentially using the braking event as a cue. Such a difference in strategy is further consistent with evidence showing that older drivers adjust their driving behaviour to compensate for slowed RTs [91]. Such adjustments to behaviour were also observed in the current study in participants’ driving speeds (Appendix A). Additionally, the posterior distribution of group differences in theta power in response to the road sign (see Figure 5; theta modulation was greater in younger than older drivers) could further corroborate a visual attention deficit in older drivers, resulting in slowed RTs and the urge to slow down. Such an interpretation is further corroborated by recent MEG findings [90] using the Callaghan, Holland and Kessler [7] paradigm.

The hypothesis that there would be age group differences in alpha modulation was qualitatively supported, although group differences did not reach significance. As shown in Figure 3, Figure 4 and Figure 5, the 18–30 years but not the 60+ years group displayed greater alpha power in the Single-Task condition compared to the Sequential-Task conditions (in both Delayed and Immediate Switch conditions). Figure 5 shows that older drivers instead display lower posterior alpha power in the Single- compared to Sequential-Task conditions. Alpha oscillations are typically thought to relate to attention, with decreased power reflecting enhanced attention [30,31,32,33,34,35,36]. The younger group therefore seem to demonstrate enhanced attention towards the road sign in the Sequential-Task compared to Single-Task conditions, which could allow for faster re-distribution of spatial attention, compatible with the RT results. Older adults, in contrast, show lower posterior alpha source power in the Single-Task condition compared to Sequential-Task conditions, which could reflect enhanced visual processing during undisturbed driving (no braking required). This interpretation is compatible with alpha effects observed during the Callaghan, Holland and Kessler [7] task, using MEG [90]. Enhanced attention to the road sign in the Sequential-Task (compared to Single-Task) conditions in the younger but not older group further supports the notion that younger drivers adjust well and quickly to dynamic changes in the driving environment, enabling them to drive at higher speeds, while older adults drive more slowly in order to cope with their reduced attentional flexibility.

### 5.3. Limitations and Future Work

There were a number of limitations to the study. Firstly, we only investigated conditions where a temporal event preceded a spatial event, and not vice versa, in order to remain consistent with Callaghan, Holland and Kessler [7] and to contain the number of conditions. This means that only inferences about switching from temporal to spatial attention can be drawn. Due to the increased length of naturalistic trials, including additional conditions was not feasible. Further research is needed to explore whether similar age-related changes are present when switching from a spatial to a temporal attention task. Based on previous literature showing that older adults find it more difficult to narrow their focus of attention from two RSVP streams to one [92], it could be expected that older adults also find it more difficult to switch from distributing their attention spatially to focusing their attention on temporally changing events (in a single location). However, it could also be that the salience of such temporal events efficiently attracts attention exogenously, outweighing increased cognitive demands.

Secondly, due to the STISIM drive software providing only the option to programme events into the scenario in distance travelled rather than time elapsed, the sign appeared either 3.04 m or 30.48 m after the braking event. This resulted in the temporal dynamics of each trial being affected by the speed that the participant was driving. This has been taken into careful consideration when interpreting findings. It further emphasises the difficulties that researchers encounter when leaving the “safety” of traditional computer experiments and embark on using more naturalistic, dynamic scenarios.

Similarly, to maintain ecological validity, participants were given control over the driving simulator vehicle and were therefore free to brake and accelerate at any time. Although the braking event did not occur until either ~0.14 s prior to the onset of the sign in the Immediate Switch condition or ~1.42 s prior to the onset of the sign in the Delayed Switch condition it is likely the participant detected the vehicle as a possible hazard prior to these time points. Older drivers have been shown to adjust their driving behaviour to compensate for slowed RTs [91] and may therefore have begun to brake at the earliest sign of a possible hazard, affecting measures of braking RT. This would explain the unusual pattern of braking RTs observed across age groups (Appendix A), where the youngest age group display slower braking RTs than older groups. Although the individual group comparisons did not reach significance with a conservative Bonferroni multiple comparisons correction, there was a significant main effect of age (*p* = 0.049). A possible explanation for the interesting pattern in braking RTs could be that younger drivers also tend to drive at higher velocities (see Appendix A), which could inflate their RTs as stimuli are changing more rapidly in the environment. An alternative explanation is that younger participants have less overall driving experience compared to older drivers. More experienced drivers conduct more anticipatory braking responses, whatever their age, e.g., [93]. See also Bao and Boyle [94] for a similar pattern of driving behaviour results, where the middle-aged group spent more time scanning their rear-view mirror compared to both younger and older adults.

A further limitation of the study is that the visual search road sign (displayed in Figure 1) contained three columns of names that the target could occur in, with the left-hand column corresponding to a left indicator response and the two right-hand columns corresponding to a right indicator response. On 50% of trials a right-hand indicator response was required and on 50% of trials a left-hand indicator response was required. This enabled participants to apply a strategy of only reading words in the left column. It is not expected that this affected our main findings for two reasons. Firstly, the number of occurrences of the target word in each column of the sign was matched across conditions, trials were pseudo-randomised within each of the three blocks and the order that the three blocks were completed in was counterbalanced across participants. Therefore, this strategy could not have been systematically implemented in one condition more than the others. Secondly, there is no evidence to suggest that older participants would be less able to identify and implement this strategy compared to younger adults. Conversely, evidence consistently shows that older participants utilise top-down cues more than younger adults, including contextual cues in a realistic visual search [83]. Older age groups displayed slower RTs and higher Dual-Task Costs compared to younger adults, suggesting that, if they were more likely to implement such a strategy, it did not affect our findings.

A common challenge in ageing research is self-selection bias during recruitment, in which volunteers tend to be highly educated, physically and socially active individuals; traits which have all been identified to improve cognitive reserve and protect against cognitive decline [95,96,97,98]. Similarly, 75% of the ARCHA participation panel members attending our previous experiments had degree level education or higher, with only three participants from the two older groups (60–69 and 70+ years groups) in Callaghan, Holland and Kessler [7] having lower than A-level equivalent education. A limitation of the current work is that we did not include level of education, physical activity or social activity as covariates in our current analysis. However, as the majority of participants in all age groups are expected to have degree level education or higher, we do not believe level of education can account for group differences in switching performance. On the contrary, it is possible that the current findings underestimate difficulties in switching in the general ageing population.

A further limitation to the current work is that groups were not matched on gender due to the practical challenges of recruiting a large sample of middle and older aged adults. We have no reason to expect that there would be gender differences in the current study. For example, Bao and Boyle [94] found no gender differences in a similar driving simulator task in which the time that participants spent visually scanning their environment was recorded. However, further research is required that directly investigates gender differences to confirm this speculation.

Lastly, the interaction between event condition and age did not reach significance when comparing the EEG subsets of participants (see Appendix A). This is likely due to the small number of participants (17 per group) and high variability (see Table 2). However, the overall pattern of indicator RTs remained the same as the data from all 116 participants (see Figure 2). Future research should to aim to replicate the current EEG findings with larger samples. Such limitations highlight the challenges of measuring brain signals in naturalistic environments. The current work presents an important step in progressing the field of naturalistic neurocognitive ageing research.

## 6. Conclusions

Virtual reality is an emerging new tool in psychological research that enables fully controlled computerised experiments with significantly increased realism. In contrast to field studies in the real-world, experimental control remains largely with the researchers and any risks imposed by real-world situations (e.g., traffic) are only virtual. Furthermore, we propose that when used in conjunction with EEG and other neuroimaging tools, unprecedented insights could be gained into human processing in realistic hazardous scenarios.

The aim of the current study was to investigate whether age-related difficulties in switching from a temporal to a spatial attention task [7] are also seen in a naturalistic setting, during simulated driving, and whether differences in behavioural performance would map onto neural oscillatory signatures. Consistent with hypotheses, RTs were slower in the two Sequential-Task conditions compared to the Single-Task condition. Unexpectedly, RTs were slower in the Delayed Switch condition compared to the Immediate Switch condition. This is likely to be due to the two well-known phenomena of increased RTs with increased time elapsed between the presentation of a cue (i.e., the braking event) and a target stimulus (i.e., the road sign) and increased RTs when the temporal onset of a target stimulus is unpredictable [50,78,79,80]. Future (hypothesis driven) work should aim to confirm this speculation.

The hypothesis that there would be an age-related decline in switching from temporal to spatial attention while driving was supported, reflected in greater Sequential-Task Costs in older age groups compared to the youngest age group. These findings support that age-related declines in refocusing from a temporal event to spatially distributed stimuli are not only observed in the computer-based switching task but are also present in naturalistic settings such as when driving. Age-related changes in task-related theta and alpha modulations were found, where older adults showed a weaker theta and alpha modulation compared to younger adults, which may imply that age groups implement different strategies to cope with attentional demands while driving.

Our findings provide a focus for the future development of training interventions to help older drivers to drive safely for longer. Driving cessation is detrimental to an older person’s independence and mental health [99,100,101]. Overall, older drivers compensate for slower RTs and general cognitive decline in their driving behaviour [91]. However, recent findings have shown that protective factors against cognitive decline in older age, such as level of education and engagement, no longer facilitate compensation later in older age, when the cognitive reserve available to compensate is no longer sufficient for the demand for compensation, caused by declines in underlying cognitive resources [102,103]. The development of an assessment and training intervention that aims to assess and improve the ability to switch between tasks during driving, such as attending to traffic and reading road signs, could therefore be beneficial to the ageing population.

## Figures and Tables

**Figure 1 brainsci-10-00530-f001:**
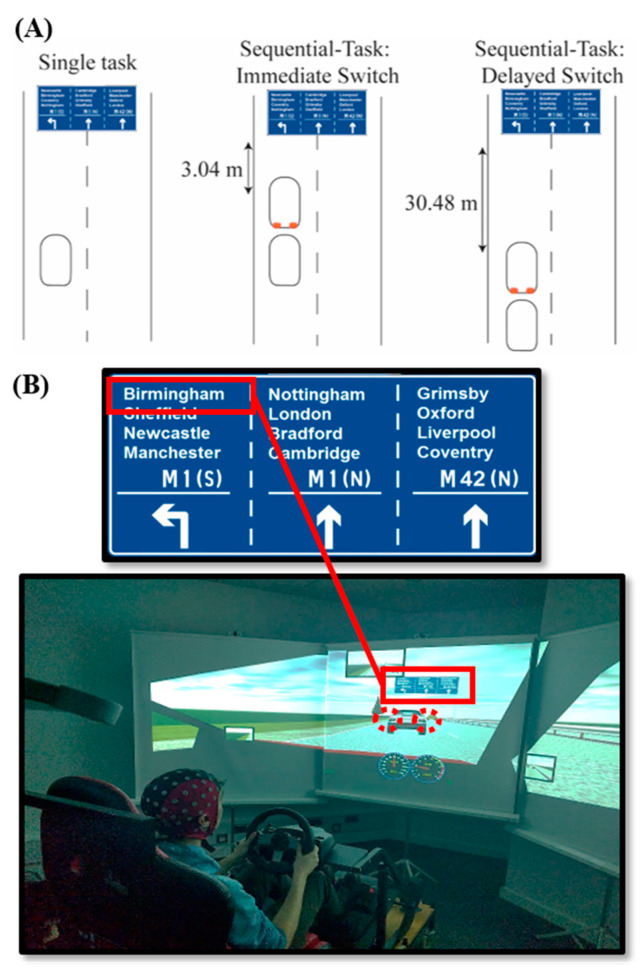
(**A**) Illustration of the three task conditions, including (from left to right) Single-Task, Sequential-Task Immediate Switch and Sequential-Task Delayed Switch. Curved rectangles represent the simulated vehicles of the participant (plain) and in the braking event (with red brake lights illustrated); (**B**) Example of the experimental setup, where the participant is seated in the driving simulator wearing an EEG cap. The projector screen displays the vehicle involved in the “braking event” (1st task; dashed red circles emphasize the brake lights), along with the road sign (2nd task, which always appeared after the braking event) during a Sequential-Task trial. As displayed enlarged at the top, the target word “Birmingham” is listed in the left column of the sign, requiring a speeded left indicator response (the location of the target on the sign varied across trials).

**Figure 2 brainsci-10-00530-f002:**
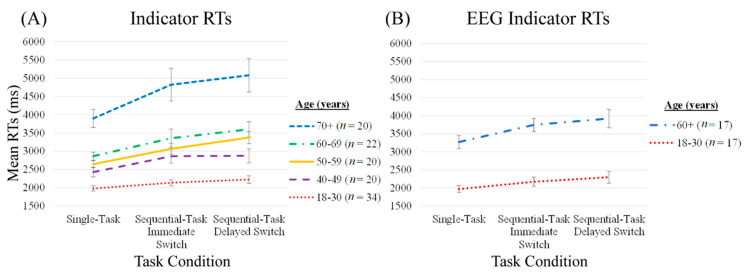
Group means of participants’ median indicator RTs in all participants (**A**) and in the subset of participants with EEG recordings (**B**). Note that the data subset shown in B is included in the overall data shown in A and in all analyses conducted with this data. Vertical bars represent the standard error of the mean.

**Figure 3 brainsci-10-00530-f003:**
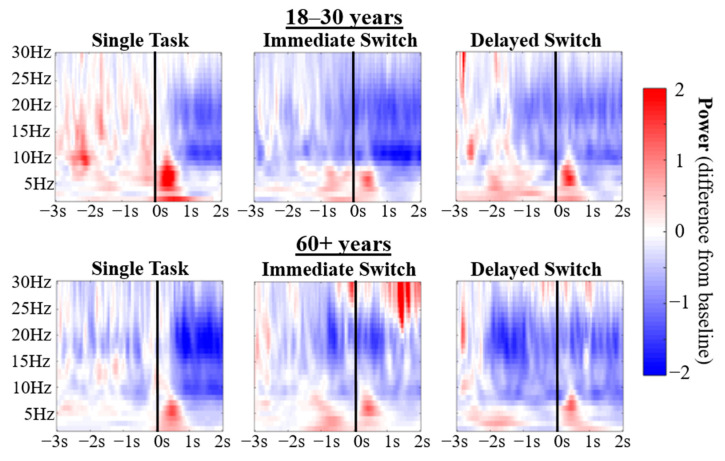
TFRs present power difference from a baseline period of −5.50 s – −3.50 s averaged across a group of 12 anterior electrodes (AF3, AF4, F1, F2, F3, F4, FC1, FC2, FC3, FC4, FC5, FC6). For an average across posterior electrodes see Appendix A and across all electrodes see Appendix A. Black lines placed over TFRs signify the onset of the road sign at 0.00 s. In the Sequential-Task conditions the car pulled in front of the participant at either 3.04 m (~0.14 s; Immediate Switch) or 30.48 m (~1.42 s; Delayed Switch) prior to the onset of the road sign.

**Figure 4 brainsci-10-00530-f004:**
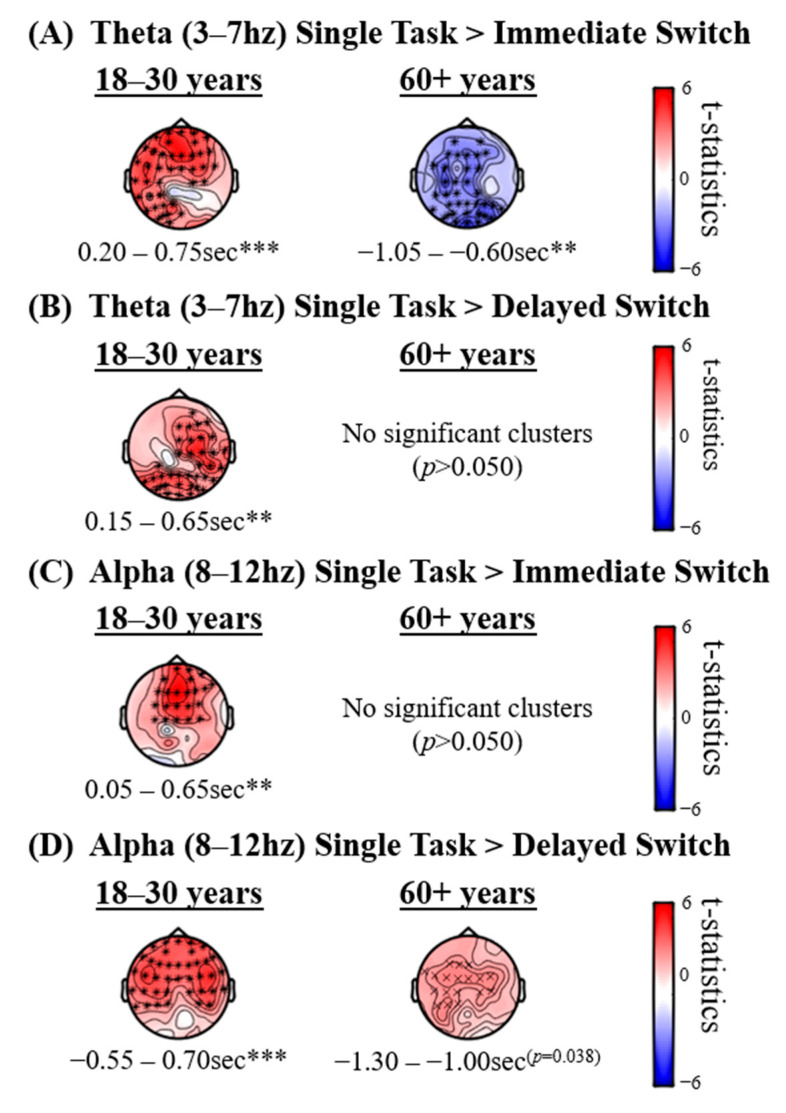
Effects in theta (3–7 Hz) and alpha (8–12 Hz) EEG power when contrasting Immediate Switch and Single-Task conditions in each age group (**A**,**C**), and contrasting Delayed Switch and Single-Task conditions in each age group (**B**,**D**). Topographical plots present *t*-statistics. Cluster significance levels from a two-tailed test are indicated as ** *p* < 0.01, *** *p* < 0.001.

**Figure 5 brainsci-10-00530-f005:**
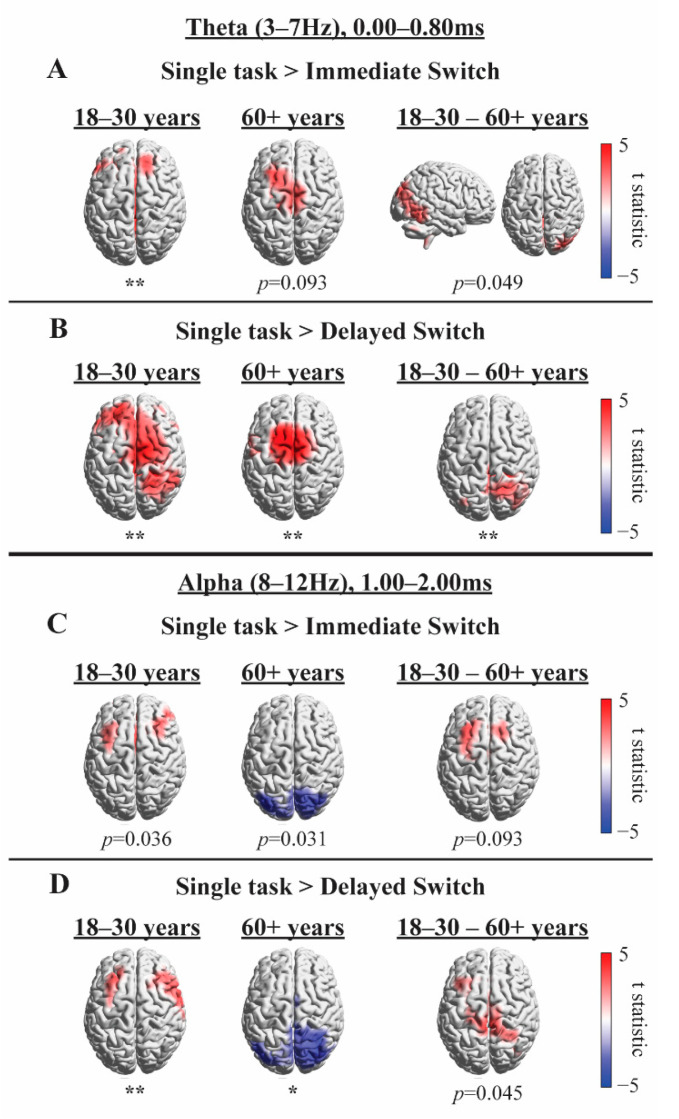
eLORETA source solutions for each age group (left and middle column) and for between-group comparisons (right column). Panels **A** and **B** depict solutions for theta (3–7 Hz; 0.00–0.80 s) and Panels **C** and **D** for alpha (8–12 Hz; 1.0–2.0 s), each for a time period after the onset of the road sign. Panels **A** and **C** show results for the Immediate Switch condition and Panels **B** and **D** for the Delayed Switch condition; both in relation to the Single-Task condition. Cluster significance levels from a two-tailed test are indicated as * *p* < 0.025, ** *p* < 0.01.

**Table 1 brainsci-10-00530-t001:** Participant demographics.

				Age Group (Years)			EEG Group	
		18–30 (*n* = 34)	40–49 (*n* = 20)	50–59 (*n* = 20)	60–69 (*n* = 22)	70–91 (*n* = 20)	18–30 (*n* = 17)	60+ (*n* = 17)
Age (years)	Mean	21.21	43.65	54.75	64.77	75.35	22.88	70.12
	SD	3.36	2.93	2.40	2.86	4.40	4.04	5.20
Sex	Male	10	8	6	14	9	4	10
	Female	24	12	14	8	11	13	7
Handedness	Right	30	17	18	20	20	15	16
	Left	4	2	2	2	0	2	1
ACE-3	Mean	N/A	N/A	96.60	96.18	95.21	N/A	95.29
	SD	N/A	N/A	2.50	2.44	2.49	N/A	2.64

The mean age of each age group, the number of male/female participants, and the number of left/right handed participants for each age group. Mean ACE-3 scores are presented for the 50–59, 60–69, and 70–91 years groups. Handedness data is missing for one participant in the 40–49 years group. Electroencephalography (EEG); Addenbrookes Cognitive Examination 3 (ACE-3); not applicable (N/A); number of participants (*n*); standard deviation (SD).

**Table 2 brainsci-10-00530-t002:** Means and SDs of Sequential-Task Costs for each age group.

	Age Group (Years)						EEG subgroup	
		18–30 (*n* = 34)	40–49 (*n* = 20)	50–59 (*n* = 20)	60–69 (*n* = 22)	70–91 (*n* = 20)	18–30 (*n* = 17)	60+ (*n* = 17)
Immediate S-T Costs	Mean	8.62	17.68	15.21	18.71	22.76	10.38	16.98
	SD	16.50	21.34	11.56	29.34	24.35	15.55	15.89
Delayed S-T Costs	Mean	12.67	18.59	26.81	27.58	30.50	15.97	21.45
	SD	20.30	20.31	14.31	24.28	32.37	23.72	21.17

Sequential-Task (S-T) Costs were calculated as the percentage increase in RT from the Single-Task condition to each of the Sequential-Task conditions (Immediate Switch/Delayed Switch) separately.

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
