# Peer review of "Age-Related Changes in Attentional Refocusing during Simulated Driving"

_brainsci, 2020, doi:10.3390/brainsci10080530_

Round 1
Reviewer 1 Report
Summary: The current manuscript investigated the age-related changes in attentional refocusing during 2 simulated driving.
The authors show age-related impairments in attentional refocusing, evidenced by increased switch-costs in response 24 times and by deficient modulation of theta and alpha frequencies.
Although authors present interesting findings and I would read the final manuscripts, some aspects could be improved.
Introduction: Overall, the introduction provides a broad background and rationale for the research. However, excessive length and some repetitive aspects may confuse the reader. I suggest a reduction.
Methods: The method is succinct and comprehensive. It could be useful justify the age range choose.
Analysis: the analyses are well conducted.
Results: The summary of the study provided is well-defined and fits according to the analysis plan provided.
Conclusion: The conclusions appear to be a summary of the results, I suggest reporting the usefulness of this study and further perspective.
General comment: I would also encourage the authors to check all references and to proofread the manuscript to improve the English language.
Author Response
Response to Reviewer 1
Firstly, we would like to thank the Reviewer for the time and effort they have invested into reviewing our manuscript and for their fast and constructive response. Please find below a point-by-point reply to each of their comments. Note that changes in reply to both Reviewers are highlighted in yellow in the manuscript.
Summary: The current manuscript investigated the age-related changes in attentional refocusing during simulated driving.
The authors show age-related impairments in attentional refocusing, evidenced by increased switch-costs in response 24 times and by deficient modulation of theta and alpha frequencies.
Although authors present interesting findings and I would read the final manuscripts, some aspects could be improved.
Introduction: Overall, the introduction provides a broad background and rationale for the research. However, excessive length and some repetitive aspects may confuse the reader. I suggest a reduction.
Reply:
Firstly, we would like to thank the Reviewer for their positive comments. We have now made changes throughout the Introduction to decrease repetitiveness and shorten the length, reducing this section by almost a third from 1430 words to 1020. We believe that these changes have made the aims of the study clearer for the reader and have improved readability, in accord with the Reviewer’s suggestion. (Added/altered text is highlighted in yellow, but deletions are not tracked)
Methods: The method is succinct and comprehensive. It could be useful justify the age range choose.
Reply:
Due to the challenges of recruiting a large sample of middle and older aged drivers, in addition to the attrition rates of older adults in driving simulator research (due to simulator sickness), we did not place an upper age limit on the 70+ years group. We have explained the age ranges on p 3 of the manuscript, now in more detail than previously:
“Consistent with Callaghan, Holland and Kessler [7], the 18-30 years group were used as a comparison group for age-related cognitive changes for all other groups. The 40-49 and 50-59 years groups provided middle-aged comparison groups for the 60-69 and 70-91 years groups. Age ranges were selected to assess performance not only in older adults, but also in middle age.” (p 3)
Analysis: the analyses are well conducted.
Results: The summary of the study provided is well-defined and fits according to the analysis plan provided.
Reply:
We would like to thank the Reviewer for their positive comments.
Conclusion: The conclusions appear to be a summary of the results, I suggest reporting the usefulness of this study and further perspective.
Reply:
We would like to thank the Reviewer for their suggestion. We have now further discussed the usefulness of the study and future perspectives in the conclusion. We believe that this has been a valuable addition to the manuscript:
“Our findings provide a focus for the future development of training interventions to help older drivers to drive safely for longer. Driving cessation is detrimental to an older person’s independence and mental health [98-100]. Overall, older drivers compensate for slower RTs and general cognitive decline in their driving behaviour [90]. However, recent findings have shown that protective factors against cognitive decline in older age, such as level of education and engagement, no longer facilitate compensation later in older age, when there are no longer cognitive resources available to compensate [101,102]. The development of an assessment and training intervention that aims to assess and improve the ability to switch between tasks during driving, such as attending to traffic and reading road signs, could therefore be beneficial to the ageing population.” (p 20)
General comment: I would also encourage the authors to check all references and to proofread the manuscript to improve the English language.
Reply:
We have now thoroughly proofread the manuscript and made some changes to improve the readability of the manuscript throughout. Note that our manuscript complies with UK English rules. Given that our 1st and 3rd author are both English native speakers (UK) we are confident that our standard of English will be deemed appropriate.
Reviewer 2 Report
The authors simulated a driving experience and assessed age differences in terms of switching-Task Costs and their electrophysiological signatures. I found the paper interesting with some missing information.
Participants. How the number of participants for both the behavioural and electrophysiological study was determined?
Education is a fundamental demographic feature for the neuropsychological assessment and, as a proxy of cognitive reserve, it influences all attentional and executive functions (e.g., Lola Roldán-Tapia, Juan García, Rosa Cánovas & Irene León (2012) Cognitive Reserve, Age, and Their Relation to Attentional and Executive Functions, Applied Neuropsychology: Adult, 19:1, 2-8).
The authors should, not only add the participants’ education in the tables but include it as a covariate in all their analyses.
It could explain at least part of the differences between groups.
The young sample is all characterized by high education (university degree) and the same could be for university staff, but the older group is not specified.
Did the authors expect differences in shifting-task costs between genders? Did they look at them (ones controlled for education)?
In “Single-Task” trials at which distance the sign appeared? This information is missing.
“On 50% of trials the target was in the left column and on 50% of the trials the target was in one of the two right hand columns.” Why the target was not presented 1/3 of the times on the left, 1/3 on the right, and 1/3 in the middle? The choice to use 50% on the left and 50% the other two could have induced a strategic exploratory behavior: looking always on the left would have been enough to decide whether to turn left or going straight ahead. On the other hand, there could be very different RTs between individuals who used those strategies and who did not and between sign positions.
Results.
Did you analyze the differences in behavior between the two age subgroups for the EEG study? There was any statistical difference in terms of Immediate or Delayed Sequential-Task Costs?
Figure S1.“A significant effect of age was found on braking RTs (F(4, 115)=2.47, p=.049). Braking RTs were faster in the 50-59 years group compared to the 18-30 years group (p=.078), however, this did not reach significance. There were no other significant age group differences in braking RTs (p>.10).” How it can be? There is an age effect but not differences among different age groups? Moreover, the trend is increasing and then decreasing, which is odd. Are young individuals on average slower than the over 70s? Difficult to believe. Could it be the result of an interaction with speed?
Discussion.
The discussion should report the similarities and the differences in respect to previous studies (e.g., Callaghan et al., 2017; and Callaghan, 2018).
Author Response
Firstly, we would like to thank the Reviewer for the time and effort they have invested into reviewing our manuscript and for their fast and constructive response. Please find below a point-by-point reply to each of their comments (we have numbered the Reviewer's comments to facilitate reading). Note that changes in reply to both Reviewers are highlighted in yellow in the manuscript.
1) The authors simulated a driving experience and assessed age differences in terms of switching-Task Costs and their electrophysiological signatures. I found the paper interesting with some missing information.
Participants. How the number of participants for both the behavioural and electrophysiological study was determined?
Reply:
A G*Power calculation revealed that a sample of 110 participants was required for detecting a medium effect size (f=0.35) with 0.9 statistical power. (We have now included this sentence in Methods, p 3).
The mobile EEG system was only available to the researchers for a limited period of time, and so the EEG samples were smaller than we would have liked (with only 17 participants in each group). We regard this sample size as exploratory (as indicated in the text, p 2), but given the applied robust statistics (cluster-based permutations with 2000 random permutations), the obtained results are reliable. We have stated:
“Electroencephalography (EEG) was recorded in a subset of 17 participants aged 18-30 and 17 aged 60+ years as a proof-of-principle pilot investigation.” (p 2)
2) Education is a fundamental demographic feature for the neuropsychological assessment and, as a proxy of cognitive reserve, it influences all attentional and executive functions (e.g., Lola Roldán-Tapia, Juan García, Rosa Cánovas & Irene León (2012) Cognitive Reserve, Age, and Their Relation to Attentional and Executive Functions, Applied Neuropsychology: Adult, 19:1, 2-8).
The authors should, not only add the participants’ education in the tables but include it as a covariate in all their analyses.
It could explain at least part of the differences between groups.
The young sample is all characterized by high education (university degree) and the same could be for university staff, but the older group is not specified.
Reply:
Education levels are indeed a relevant factor in research on age-related decline, and we would like to thank the Reviewer for their suggestion. While we agree that level of education could have influenced task performance in our specific sample, we believe that it would do so to lessen group differences rather than increase them. A limitation of the current participant sample is the common self-selection bias, in which our research volunteers tend to be highly educated, highly motivated, healthy individuals, which we have reported in previous research (see below). We therefore argue that group differences in level of education could not account for our effects, but rather cause us to underestimate group differences in task performance in the general population. We have now dedicated a section of the Discussion to this:
“A common challenge in ageing research is self-selection bias, in which volunteers tend to be highly educated, physically and socially active individuals, traits which have all been identified to improve cognitive reserve and protect against cognitive decline [94-97]. Similarly, 75% of the ARCHA participation panel members attending our previous experiments had degree level education or higher, with only three participants from the two older groups (60-69 and 70+ years groups) in Callaghan, et al. [7] having lower than A-level equivalent education. A limitation of the current work is that we did not include level of education, physical activity or social activity as covariates in our current analysis. However, as the majority of participants in all age groups have degree level education or higher, we do not believe level of education can account for group differences in switching performance. On the contrary, it is possible that the current findings underestimate difficulties in switching in the general ageing population.” (pp 18-19)
3) Did the authors expect differences in shifting-task costs between genders? Did they look at them (ones controlled for education)?
Reply:
We fully agree with the Reviewer that gender is an important factor to consider in ageing research. However, due to the challenges of recruiting such a large sample of adults across middle age and older age groups (total n=128; 12 excluded), it was not feasible to match groups on gender, or recruit sufficient numbers to include it as a further factor in the analysis. We therefore do not have sufficient participant numbers to conduct an adequate investigation of gender differences. Previous work has found an absence of gender differences in a similar driving simulator task, which similarly involved participants visually scanning their environment [6]. We therefore have no reason to expect gender differences in the current paradigm, however further research with appropriately large and gender-balanced samples is required to confirm this. We have now incorporated this as a limitation in the discussion on p 19:
“A further limitation to the current work is that groups were not matched on gender due to the practical challenges of recruiting a large sample of middle and older aged adults. We have no reason to expect that there would be gender differences in the current study. For example, Bao and Boyle [93] found no gender differences in a similar driving simulator task in which the time that participants spent visually scanning their environment was recorded. However, further research is required that directly investigates gender differences to confirm this speculation.”
4) In “Single-Task” trials at which distance the sign appeared? This information is missing.
Reply:
As the paradigm was a naturalistic driving simulator study, the “trials” ran on continuously from one another, with certain events occurring within the ongoing driving scenario being considered as trials. Each trial was therefore time-locked to the onset of the road sign, as is evident in the TFRs in Figure 3. The distance between road signs varied so that participants could not predict when the sign would appear. We have now better explained the continuous and dynamic aspect of our paradigm in the Introduction (p 2) Methods (p 5).
5) “On 50% of trials the target was in the left column and on 50% of the trials the target was in one of the two right hand columns.” Why the target was not presented 1/3 of the times on the left, 1/3 on the right, and 1/3 in the middle? The choice to use 50% on the left and 50% the other two could have induced a strategic exploratory behaviour: looking always on the left would have been enough to decide whether to turn left or going straight ahead. On the other hand, there could be very different RTs between individuals who used those strategies and who did not and between sign positions. (p 10)
Reply:
Participants were required to respond to the road sign with a left indicator response if the target name “Birmingham” was in the left column and make a right indicator response if the target was in either of the two right hand columns. The target was therefore in the left column 50% of trials and in one of the two right hand columns 50% of the trials so that participants could not predict the probability of whether a left/right indicator response would be required. We have now made this more explicit with the following addition on p 6:
“On 50% of trials the target was in the left column and on 50% of the trials the target was in one of the two right hand columns, so that 50% of trials required a left hand indicator response and 50% of trials required a right hand indicator response (see above for details).”
While we agree that this strategy of searching only the left column was possible, we do not believe that this affected our main pattern of results. It is unlikely that younger age groups were more likely to implement such a strategy compared to older age groups, which we have now discussed in the manuscript:
“A further limitation of the study is that the visual search road sign (displayed in Figure 1) contained three columns of names that the target could occur in, with the left-hand column corresponding to a left indicator response and the two right-hand columns corresponding to a right indicator response. On 50% of trials a right-hand indicator response was required and on 50% of trials a left-hand indicator response was required. This enabled participants to apply a strategy of only reading words in the left column. It is not expected that this affected our main findings for two reasons. Firstly, the number of occurrences of the target word in each column of the sign was matched across conditions, trials were pseudo-randomised within each of the three blocks and the order that the three blocks were completed in was counterbalanced across participants. Therefore this strategy could not have been systematically implemented in one condition more than the others. Secondly, there is no evidence to suggest that older participants would be less able to identify and implement this strategy compared to younger adults. Conversely, evidence consistently shows that older participants utilise top-down cues more than younger adults, including contextual cues in a realistic visual search [83]. Older age groups displayed slower RTs and higher Dual-Task Costs compared to younger adults, suggesting that, if they were more likely to implement such a strategy, it did not affect our findings.” (pp 18)
6) Did you analyze the differences in behavior between the two age subgroups for the EEG study? There was any statistical difference in terms of Immediate or Delayed Sequential-Task Costs?
Reply:
We have now included an analysis of the EEG subgroups’ indicator RTs in the supplementary material, along with a discussion of findings in the Discussion on p 19. As the 34 participants in the EEG driving groups are included in the main analysis of all 116 participants, we have not repeated the analysis of this data in the main text, as this could be misleading. In short, variability in RTs and smaller N in the EEG subgroup prevented Switch-Costs to reach significance, but the pattern is in the identical direction as for the overall group analysis (see Figure 2).
7) Figure S1. “A significant effect of age was found on braking RTs (F(4, 115)=2.47, p=.049). Braking RTs were faster in the 50-59 years group compared to the 18-30 years group (p=.078), however, this did not reach significance. There were no other significant age group differences in braking RTs (p>.10).” How it can be? There is an age effect but not differences among different age groups? Moreover, the trend is increasing and then decreasing, which is odd. Are young individuals on average slower than the over 70s? Difficult to believe. Could it be the result of an interaction with speed?
Reply:
We agree that the pattern of braking RTs was surprising. The main effect of age was just below threshold (p=.049). It is likely that individual group differences did not survive the conservative Bonferroni multiple comparisons correction. The Reviewer could indeed be correct and higher velocities in younger participants, when the car in front is braking, could lead to longer RTs. We have now added to our discussion of the braking RTs in the manuscript:
“Older drivers have been shown to adjust their driving behaviour to compensate for slowed RTs [90] and may therefore have begun to brake at the earliest sign of a possible hazard, affecting measures of braking RT. This would explain the unusual pattern of braking RTs observed across age groups (Figure S1), where the youngest age group display slower braking RTs than older groups. Although the individual group comparisons did not reach significance with a conservative Bonferroni multiple comparisons correction, there was a significant main effect of age (p=.049). A possible explanation for the interesting pattern in braking RTs could be that younger drivers also tend to drive at higher velocities (see Figure S2), which could inflate their RTs as stimuli are changing more rapidly in the environment. An alternative explanation is that younger participants have less overall driving experience compared to older drivers. More experienced drivers conduct more anticipatory braking responses, whatever their age [e.g. 92]. See also Bao and Boyle [93] for a similar pattern of driving behaviour results, where the middle-aged group spent more time scanning their rear-view mirror compared to both younger and older adults.” (p 18)
8) Discussion.
The discussion should report the similarities and the differences in respect to previous studies (e.g., Callaghan et al., 2017; and Callaghan, 2018).
Reply:
We would like to thank the Reviewer for the suggestion to discuss the similarities and differences in relation to our previous work. We have now incorporated a comparison throughout the Discussion (for example on p 15 and p 17 of the manuscript).
Round 2
Reviewer 2 Report
I found the authors answered all my concerns. I think that the manuscript is clearer and deserves publication.